# Evaluation of Marine Diindolinonepyrane in Vitro and in Vivo: Permeability Characterization in Caco-2 Cells Monolayer and Pharmacokinetic Properties in Beagle Dogs

**DOI:** 10.3390/md17120651

**Published:** 2019-11-20

**Authors:** Zibin Ma, Ruihua Guo, Jeevithan Elango, Bin Bao, Wenhui Wu

**Affiliations:** 1College of Food Science and Technology, Shanghai Ocean University, Shanghai 201306, China; zibinma@163.com (Z.M.); rhguo@shou.edu.cn (R.G.); srijeevithan@gmail.com (J.E.); 2Shanghai Engineering Research Center of Aquatic-Product Processing & Preservation, Shanghai 201306, China

**Keywords:** fibrinolytic compound, Caco-2 cell, apparent permeability coefficients (P_app_), dogs, pharmacokinetics

## Abstract

A marine fibrinolytic compound was studied for use in thrombolytic therapy. Firstly, the absorption and transportation characteristics of 2,5-B*H*PA (2,5-B*H*PA:2,5-Bis-[8-(4,8-dimethyl-nona-3,7-dienyl)-5,7-dihydroxy-8-methyl-3-keto-1,2,7,8-tertahydro-6*H*-pyran[a]isoindol-2-yl]-pentanoic acid, a novel pyran-isoindolone derivative with bioactivity isolated from a rare marine microorganism in our laboratory) in the human Caco-2 cells monolayer model were investigated. We collected 2,5-B*H*PA in the cells to calculate the total recovery, and its concentration was analyzed by LC/MS/MS (Liquid Chromatography/Mass Spectrum/Mass Spectrum). The results showed that 2,5-B*H*PA has low permeability and low total recoveries in the Caco-2 cells membrane. Pharmacokinetics and tissue distribution of 2,5-B*H*PA were investigated in beagle dogs using HPLC (High Performance Liquid Chromatography) after intravenous administration of three different doses (7.5, 5.0, 2.5 mg·kg^−1^). Pharmacokinetic data indicated that 2,5-B*H*PA fitted well to a two-compartment model. Elimination half-lives (T_1/2_) were 49 ± 2, 48 ± 2, and 49 ± 2 min, respectively; the peak concentrations (C_max_) were 56.48 ± 6.23, 48.63 ± 5.53, and 13.64 ± 2.76 μg·mL^−1^, respectively; clearance rates (CL) were 0.0062 ± 0.0004, 0.0071 ± 0.0008, and 0.0092 ±0.0006 L·min^−1^·kg^−1^, respectively; mean retention times (MRT) were 28.17 ± 1.16, 26.23 ± 0.35, and 28.66 ± 0.84 min, respectively. The low penetrability of 2,5-B*H*PA indicated that the intravenous route of administration is more appropriate than the oral route. Meanwhile, 2,5-B*H*PA showed a good pharmacokinetic profile in beagle dogs. The tissue distribution showed that 2,5-B*H*PA could quickly distribute into the heart, intestines, liver, stomach, spleen, lungs, testicles, urine, intestine, kidneys, brain, and feces. The concentration of 2,5-B*H*PA was higher in the liver and bile. Interestingly, 2,5-B*H*PA was detected in the brain. Taken together, the above results suggested that our work might be beneficial in the development of agents for thrombolytic treatment.

## 1. Introduction

Thrombosis is major human disease, and ~40% of deaths globally are due to cardiovascular disease [1,2,3]. The Caco-2 cell line was derived from human colon adenocarcinoma cell lines. Its morphology, functional expression of marker enzymes, formation of multiple active transport systems, and infiltration characteristics are similar to those of intestinal epithelial cells [4]. According to the literature, the Caco-2 cell line has become an effective tool for the study of drug uptake, excretion, trans-cellular transport, and other absorption mechanisms [5,6], and has become an indispensable method for the study of drug absorption. Therefore, the Caco-2 cell model can be used as an in vitro model to study the transport and metabolism of drugs in small intestinal epithelial cells, and is applicable to the early stage of new drug development to study the absorption and transport of drugs [7]. Rouquayrol et al. [8] used the Caco-2 cell model to improve the oral bioavailability of various precursors of HIV protease inhibitors, namely saquinavir, indinavir, and nelfinavir.

Previously, 2,5-B*H*PA (2,5-Bis-[8-(4,8-dimethyl-nona-3,7-dienyl)-5,7-dihydroxy-8-methyl-3-keto-1,2,7,8-tertahydro-6*H*-pyran[a]isoindol-2-yl]-pentanoic acid; Figure 1) has been shown to have fibrinolytic activity [9]. It was isolated from a rare marine fungi *Stachybotrys longispora* FG216 [10]. The structure of 2,5-B*H*PA has been elucidated by Guo et al. [11]. 2,5-B*H*PA activates fibrinolytic enzyme activity, which decreases thrombolysis while minimizing bleeding risk [11,12,13].

2,5-B*H*PA showed good pharmacokinetic characteristics in rats [14]. The concentration variation of 2,5-BHPA in blood with time was detected in a few minutes after administration of an intravenous (IV) bolus. The half-life of 2,5-B*H*PA was ~22 min in rats [14]. 

The objective of this study was to investigate absorption and transport characteristics of the small molecular compound 2,5-B*H*PA by assessing the effusion ratio, total recovery rate, and apparent permeability coefficients (P_app_) of 2,5-B*H*PA using the Caco-2 cell model. Moreover, we evaluated the pharmacokinetics and tissue distribution after intravenous injection in beagle dogs’ plasma and tissues using HPLC analysis [15,16].

## 2. Results

### 2.1. Establishment and Evaluation of Caco-2 Cell Model

As shown in Figure 2, after digestion and passage of Caco-2 cells, it can be observed that the cells grow densely and gradually stick to the wall. Microscope images showed uniform and round cell sizes, clear boundaries, dense cells, and clear lines of cell proliferation after 21 days of culture in the transwell plate.

The Caco-2 cell membrane resistance value changed between 247 ± 22 Ω·cm^2^. More than 200 Ω·cm^2^ cells can be used for official testing [17]; thus, theCaco-2 cell model was successfully established and could be used in the next transport experiment.

The transmembrane flux of leaky markers such as luciferin yellow can also be used as an evaluation index to determine the cell integrity of the Caco-2 cell model [18]. Under the same conditions, the P_app_ of fluorescence yellow was (0.15 ± 0.027) × 10^−6^ cm·s^−1^, significantly lower than the generally acceptable level (2.5 × 10^−6^ cm·s^−1^). The transmittance was 0.04 ± 0.005 % and lower than the acceptable level of 1%.

In conclusion, the Caco-2 cell model established presented integrity. The transmembrane resistance and fluorescence yellow transmittance of the cells all met the requirements described in the literature [19,20].

### 2.2. Analysis of Absorption and Transport Characteristics of 2,5-BHPA

The total recovery rate of control drugs fenoterol, propranolol, and digoxin was more than 83%, among which the total recovery rate of propranolol in the direction of apical side (AP) → basolateral side (BL) was 100%. By contrast, the total recovery rate of 2,5-B*H*PA fluctuated in the ranges of 35.8%–52.4% and 25.9%–53.0% in the AP→BL and BL→AP directions, respectively. 2,5-B*H*PA showed relatively low displacement ratio and total recovery rate. P_app_ ≤ 2.5 (×10^−6^ cm·s^−1^) indicated low permeability. The data are listed in Table 1.

To overcome the non-specific adsorption effect, we added 0.1% bull serum albumin (BSA) to the AP side (AP→BL) and the BL side (BL→AP) and increased the concentration to 5, 15, and 25 µM. The data are listed in Table 2. As shown in Figure 3, the compounds present in biological samples were well separated. Figure 4 shows the mass spectrum of 2,5-BHPA in positive ESI (electro spray ionization).

From Table 2, when the total recovery rate of 2,5-B*H*PA was higher than 90%, whether it was in the AP→BL or BL→AP direction, P_app_ ≤ 2.5 (×10^−6^ cm·s^−1^); thus, 2,5-B*H*PA showed low permeability. It was suggested that 2,5-B*H*PA was not a substrate of P-gp and did not participate in cell metabolism.

There was no obvious difference in P_app_ measured in the AP→BL and BL→AP directions among all treatment groups 2 µM to 25µM, and most efflux ratios exceeded 1.5. The results indicate that the transport of 2,5-B*H*PA in the Caco-2 model may involve passive diffusion.

### 2.3. Linearity of Standard Curve

The experimental results showed that the linear regression equation of 2,5-B*H*PA is Y = 0.9561X + 0.1264 (μg·mL^−1^, R^2^ = 0.9997) in the concentration range of 0.50–500 mg·L^−1^, meaning that 2,5-B*H*PA owned a good linear relationship in the concentration of 0.50–500 mg·L^−1^. 

### 2.4. Detection of 2,5-BHPA in Plasma System

Table 3 shows that the volatility of the relative standard deviation (RSD) in all plasma samples was within 7% when beagle dogs were injected with different concentrations of 2,5-B*H*PA (1.0, 100, 400 μg·mL^−1^), which indicated that 2,5-B*H*PA was stable in the preservation and treatment process.

### 2.5. Pharmacokinetics

The PKsolver software edited by China Pharmaceutical University (Nanjing, China) was used to analyze pharmacokinetics. The survival square sum (SUM), Akaike’s information criterion (AIC), and fitted degree (R^2^) were used to establish the two-compartment model (weight = 1/c^2^), which was more suitable for the plasma concentration-time curves. The plasma concentration-time curve is shown in Figure 5.

The pharmacokinetic parameters of 2,5-B*H*PA are presented in Table 4. The results of statistical analysis showed that the area under the curve (AUC) value increased significantly with the linear increase of dose. There was no significant difference in CL (clearance) and MRT (mean residence time) at each dose. According to the survival square sum (SUM), Akaike’s information criterion (AIC), and fitted degree (R^2^), the pharmacokinetic rule of 2,5-B*H*PA is consistent with the two-compartment model (w = 1/c^2^). 

### 2.6. Tissue Distribution

The concentration of 2,5-B*H*PA in different organs after 1 h of 2,5-B*H*PA injection at a dose of 7.5 mg/kg is shown in Figure 6. The distribution of 2,5-B*H*PA was widely detected in various tissues, and the concentration in the liver and bile was higher compared with other organs. Lung, kidney, spleen, stomach, excrement, and urine contained moderate amounts of 2,5-B*H*PA. However, the specific excretion pathways of 2,5-B*H*PA and the structure and activity of metabolites need to be demonstrated through more rigorous experiments.

## 3. Discussion

The Caco-2 cells monolayer model has been used frequently to study drug absorption mechanisms and predict the delivery route of lead compounds through various pathways [4,5]. The best study result is passive transmembrane transport, while active carrier-mediated transport can predict that the best delivery route is intravenous rather than oral administration [6,7,8]. The total recovery of 2,5-B*H*PA fluctuated between the AP→BL and BL→AP directions, with ranges of 35.8%–52.4% and 25.9%–53.0%, respectively. The P_app_ ranged from 0.04 × 10^−6^ to 0.49 × 10^−6^ cm·s^−1^. Low permeability and low recovery were shown compared with the three controls. After the addition of 1% BSA, the recovery rate was greatly improved, but the permeability was still low. Therefore, it was concluded that the low recovery rate was caused by cell adsorption or cell metabolism, and 2,5-B*H*PA is a low-permeability compound. Therefore, regarding the delivery route of 2,5-B*H*PA, oral administration may not be the most appropriate, and intravenous injection may be preferable. However, previous studies have shown that different laboratories differ in cell morphology, monolayer integrity, and transport characteristics due to different cell origins and heterogeneity during cell differentiation. Therefore, appropriate methods should be used to evaluate the established Caco-2 model, and the comparability of experimental data should be ensured [17]. In this study, the TEER (transepithelial electrical resistance) value was used as the integrity indicator to judge the monolayer model of Caco-2 cells. In addition, fluorescence yellow, a drug with difficult absorption and paracellular transport, and drug propranolol, a drug with easy absorption and cross-cell transport, were selected as positive control drugs, and their P_app_ was used to further evaluate the tightness and permeability of the model. The TEER value of fluorescence yellow and P_app_ of propranolol in this study were in accordance with literature reports [17]. Bisindoles were the first compounds found to be unable to penetrate Caco-2 cells [12].

A simple, reliable, and validated HPLC method was proposed for measuring 2,5-B*H*PA in dogs’ plasma and tissues. The pharmacokinetics in dogs fitted a two-compartment model, according to survival square sum (SUM), fitted degree (R^2^), and Akaike’s information criterion (AIC). Systemic clearance (CL) and mean residence time (MRT) indicated that 2,5-B*H*PA has linear pharmacokinetic characteristics in dogs, which showed no significant differences between the dose ranges tested. The half-life fits the expectations according to previous studies [14]. The data of half-life (T_1/2_) in dogs in the present study was about two times longer than in rats [14].

Tissue distribution revealed that 2,5-B*H*PA could quickly distribute into the heart, stomach, liver, urine, spleen, brain, lungs, testicles, kidneys, intestine, and feces. The highest 2,5-B*H*PA concentration level was detected in the liver, which was same as in rats [14]. Interestingly, 2,5-B*H*PA was detected in the brain of dogs, which was not the case in rats [14]. According to tissue distribution, we could detect the drug prototype 2,5-B*H*PA in dog organs such as heart, lung, and brain. After 60 min, both liver and small intestine had a higher concentration of 2,5-B*H*PA than other organs. Therefore, 2,5-B*H*PA has a long half-life (T_1/2_), indicating that it can act on various thrombosis symptoms, including cerebral thrombosis, pulmonary embolism, and myocardial infarction.

Therefore, 2,5-B*H*PA is a new marine lead compound (separated and purified from the rare marine fungus *Stachybotrys longispora* FG216) that has a novel structure, fibrinolytic activity, and low molecular weight [9,10,11,12]; our results suggest that that 2,5-B*H*PA has potential medicinal properties and can be used in the development of a thrombolytic drug candidate.

## 4. Materials and Methods 

### 4.1. Reagents and Instruments

2,5-B*H*PA (purity > 98% using HPLC) was prepared in the marine drugs laboratory at Shanghai Ocean University. The active compound 2,5-B*H*PA with fibrinolytic activity measured by micro-plate reader was isolated from a culture broth and refined with marine fungi *Stachybotrys longispora* FG216 (CCTCCM 2012272) by using semi-preparative HPLC. This material was subjected to preparative HPLC on an Inertsil PREP-ODS column (22.5 × 250 mm), which was developed at 40 °C with a gradient elution of acetonitrile and 0.1% trifluoroacetic acid at a rate of 10 mL/min. The fraction containing the fibrinolytic product was evaporated to remove acetonitrile and trifluoroacetic acid after the purified compound was extracted with ethyl acetate. Caco-2 cells were purchased from the Life Science College Cell Resource Center in Shanghai, Chinese Academy of Sciences (original cells were from America Type Culture Collection). The CO-150 INNOVA CO^2^ incubator was purchased from NBS Co., Ltd. (USA); the mass spectrometer API 4000 from Sciex (Waltham, MA, USA); the chromatographic column Boston Crest Cyano, 120 Å, 5.0 µm (2.1 × 30 mm) from Shimadzu Corp., Kyoto, Japan; the analytical column Symmetry C18 5.0 µm (4.6 × 250 mm) from Waters, Framingham, USA; the IX51 inverted microscope from Olympus (Tokyo, Japan); the Millicell-ERS transmembrane resistance meter from Millipore (MA, USA); and the thermo3001VARIOSKANFLASH from Thermo Fisher (Shanghai, China). The HPLC system (Hitachi Corp., Tokyo, Japan) consisted of a L-2400 ultraviolet detector, L-2130 pump, L-2200 auto-sampler, and D-2000 HPLC software. All analytical grade reagents were obtained from Sigma (USA). Ultrapure water was used throughout the study.

### 4.2. Analysis Conditions

The mass spectrum condition and parameters were as follows. The mobile phase had varying ratios of acetonitrile–water (0.1% formic acid) at a flow rate of 1mL/min. The column oven was maintained at 40 °C. The electro spray ionization source (ESI) was used with negative-ion mode. The range of m/z was 100–900. Capillary voltage was 3.0 kV. Taper hole voltage was 40 kV. The ion source temperature was 100 °C. The desolvation temperature was 450 °C. The velocity of dissolvent gas was 900 L·h^−1^. The cone gas flow was 50 L·h^−1^. The detection mode was multiple ion monitoring. The software API 4000 was used in the experiment. The m/z of fenoterol, propanolol, internal standard (tolbutamide), and 2,5-B*H*PA was 304.2, 270.0, 271.1, and 869.5, respectively, which were detected in positive ESI. The m/z of digoxin was 779.6, which was detected in negative ESI. 

The HPLC (Hitachi) with the symmetry column condition had varying ratios of acetonitrile–water (0.1% trifluoroacetic acid) and the gradient was from 45:55 to 85:15 (v/v) at a flow rate of 1 mL/min. The column temperature was maintained at 40 °C. Eluents were monitored at 265 nm.

### 4.3. Animals and Ethical Statement

Male beagle dogs (SPF, 10.2–10.8 kg, aged 2 years) were provided by the Xingang Experimental Animal Co., Ltd. (Shanghai, China). All dogs were fed with standard laboratory food and water ad libitum at least 3 days prior to the experiments in an environmentally controlled room (12 h dark-light cycle; ~25 °C, 60% humidity, GB14925-2001). All protocols and procedures were approved by the Institutional Animal Care and Use Committee of Shanghai Ocean University (Approval number: 13-0012). The blood and tissues of the dogs were removed while the dogs were ether anesthetized. Their plasma was collected from the fossa orbitalis vein. Every effort was made to minimize animal pain and suffering. 

### 4.4. In Vitro Experiments

#### 4.4.1. Preparation of Samples

2,5-B*H*PA was weighed and dissolved in DMSO (dimethylsulfoxide) as a solvent (5 mM for stock solutions). The stock solutions were diluted with HBSS (Hank’s balanced salt solution) (pH 7.4) to standard solutions of different concentrations. The final concentration of DMSO was less than 1%.

Tolbutamide (internal standard) was weighed precisely and dissolved in DMSO as a solvent (0.5 mM for stock solution). The stock solution was diluted to 5 μM with the fluid phase as internal standard.

#### 4.4.2. Establishment of Caco-2 Cells Monolayer Model

Caco-2 cells were cultured in a CO-150 INNOVA CO^2^ incubator with DEME (Dulbecco’s modified eagle) medium, including 10% fetal calf serum, 1% non-essential amino acid, and penicillin (streptomycin dual) resistance solution. The cells were cultured at constant temperature in a closed incubator with the following conditions: 37 °C, 5% CO^2^, and 95% relative humidity. When the cell growth reached the fusion, 0.05% trypsin and EDTA were added and digested at 37 °C for 5 min. The cells were inoculated on a 96-well transwell culture plate with an inoculation concentration of 1 × 10^5^/cm^2^ for the cell transport experiment. After inoculation, the solution was changed every 4 to 6 days, and the culture was observed by electron microscope at 21 to 28 days [6]. The TEER value was detected by a Millicell-ERS transmembrane resistance meter as the integrity indicator to judge the monolayer model of Caco-2 cells.
TEER = (R_t_ − R_0_) × A (Ω cm^2^)(1)
where R_t_ is the resistance value of Cell pore; R_0_ is the resistance value of blank; and A is the compartment effective membrane area

#### 4.4.3. Absorption and Bidirectional Transport Experiment

The samples included three control drugs (fenoterol, propranolol, and digoxin) and 2,5-B*H*PA. The Caco-2 cell model was established successfully, and the transwell plate was then rinsed with HBSS solution three times and cultured in an incubator at 37 °C for 30 min to remove the interfering substances on the cell surface. For transfer in the AP→BL direction, 0.075 mL drug solution was added to the AP side as the supply pool, and 0.25 mL HBSS solution was added to the BL side as the receiving pool. In the BL→AP direction, 0.25 mL compound solution was added to the BL side as the supply pool, and 0.075 mL HBSS solution was added to the AP side as the receiving pool. The transwell plate was then cultured in an incubator at 37 °C for 2 h. Samples on the AP side and BL side were collected, respectively. Then, acetonitrile containing an internal label was added to each upper chamber to lyse transport membranes. The total recovery was calculated by collecting the amount of lead compound absorbed on the membrane surface or accumulated in the cell. Finally, the concentration of each sample was calculated as the ratio of the peak area between the compound and the inner target through mass spectrum analysis (API 4000). 

The absorption and transport characteristics of 2,5-B*H*PA in the Caco-2 monolayer cell model were evaluated with the apparent permeability coefficient (P_app_), effusion ratio, and total recovery rate.
P_app_ = (dC_r_/dt) × V_r_/(A × C_0_)(2)
where dC_r_/dt is the cumulative compound concentration per unit of time (µM/s); V_r_ is the volume in the receiving hole (mL); A is the transport membrane area; and C_0_ is the compound initiation concentration (µM).
Efflux ratio% = P_app_ (BL → AP)/P_app_ (AP → BL)(3)
Recovery rate% = 100 × [(V_r_ × C_r_) + (V_d_ × C_d_)]/(V_d_ × C_0_)(4)
Total recovery rate% = 100 × [(V_r_ × C_r_) + (V_d_ × C_d_) + (V_c_ × C_c_)]/(V_d_ × C_0_)(5)
where V_d_ is the volume of dosing hole (mL); C_d_ and C_r_ are the dose and receiving concentrations (µm); C_c_ is the cell lysate concentration (µm); and V_c_ is the volume of acetonitrile added to the upper chamber (mL).

#### 4.4.4. Calibration Curve for 2,5-BHPA

2,5-B*H*PA was dissolved in normal saline with NaHCO_3_ as a solvent (2,5-B*H*PA/NaHCO_3_, m/m, 1:1). The stock solution was serially diluted to 0.5, 1, 10, 50, 100, 200, and 500 mg/L. All solutions were diluted with methanol [21] and filtered using a 0.22 µm disposable syringe filter before HPLC analysis. A linear regression equation was generated for the concentration of 2,5-B*H*PA (X) versus the HPLC peak area (Y).

#### 4.4.5. Detection of 2,5-BHPA in Plasma System in Vitro

The recovery rate of samples (1.0, 100.0 and 400.0 mg/L 2,5-B*H*PA) was measured, respectively. Each sample was added to quantitative methanol. After blending, the supernatant was filtered for HPLC detection after centrifugation at 5000× *g* for 15 min at room temperature (RT). The ratio of 2,5-B*H*PA extracted by methanol to 2,5-B*H*PA detected by quality control samples was defined as the recovery rate.

The stability of 2,5-B*H*PA was assayed at 1, 100, and 400 mg/L. The short-term stability of 2,5-B*H*PA stored at 37 °C was determined after 24 h. Stability was defined as the amount of 2,5-B*H*PA residual after 24 h divided by the initial amount. 

### 4.5. In Vivo Experiments

2,5-B*H*PA was dissolved in normal saline with NaHCO_3_ as a solvent (2,5-B*H*PA/NaHCO_3_, m/m, 1:1), which was agitated overnight on a magnetic stirrer. Subsequently, it was diluted to different concentrations, filtered, and stored in a refrigerator at 4 °C. The dose was adjusted according to the body weight of the beagle dogs in a volume of 200 µL, and 2,5-B*H*PA was administered by intravenous injection (7.5, 5.0, and 2.5 mg/kg). Subcutaneous injection in the forearm was used in the experiment.

Nine dogs were randomly divided into three groups (*n* = 3) and fasted for 24 h with water ad libidum. 2,5-B*H*PA was administered by intravenous injection (7.5, 5.0, and 2.5 mg/kg). The volume of the three sample levels was identical. The blood samples of 200 µL were drawn at 0, 1, 5, 15, 30, 40, 60, 120, and 240 min after drug treatment, and the samples were immediately stored with 1.8 mL of methanol (methanol:blood = 9:1, v/v). The samples were mixed and centrifuged at 5000× *g* for 15 min at RT, and the supernatants were filtered for HPLC analysis.

Moreover, another three dogs were fasted for 24 h with water ad libidum for the tissue distribution study. 2,5-B*H*PA was administered by intravenous injection at 7.5 mg/kg. Subsequently, their brain, stomach (emptied of gastric contents), heart, large intestine (emptied of gastric contents), liver, kidneys, spleen, urine, lungs, spermary, small intestine (emptied of gastric contents), feces, muscle, bile of dogs were removed at 60 min after compound administration. The tissues were flushed with normal saline, dried, weighed, and homogenized in proper methanol (methanol:sample = 5:1, V/V). The tissue samples were centrifuged at 5000× *g* for 15 min at RT, and the supernatants were filtered with a 0.22 µm disposable syringe filter for HPLC detection.

### 4.6. Data Processing

The data were processed with Microsoft Excel 2013 edited by Microsoft (Seattle, WA, USA) and SPSS V25.0 edited by SPSS (Chicago, IL, USA). PKsolver V2.0 edited by China Pharmaceutical University (Nanjing, China) was used to calculate the pharmacokinetic parameters and the compartment model [22]. The image was processed with Origin 9.0 software edited by Origin Lab (Northampton, MA, USA).

## 5. Conclusions

In this study, a marine fibrinolytic compound was studied for use in thrombolytic therapy. The results of absorption and transportation characteristic of 2,5-B*H*PA in the human Caco-2 cells monolayer model showed low permeability and low total recoveries. Low penetrability of 2,5-B*H*PA indicated that the intravenous route of administration is more appropriate than the oral route. Meanwhile, 2,5-B*H*PA showed a good pharmacokinetic profile in beagle dogs. The tissue distribution showed that 2,5-B*H*PA could quickly distribute into organs. Taken together, the above results suggested that our work might be useful for developing agents for thrombolytic treatment.

## Figures and Tables

**Figure 1 marinedrugs-17-00651-f001:**
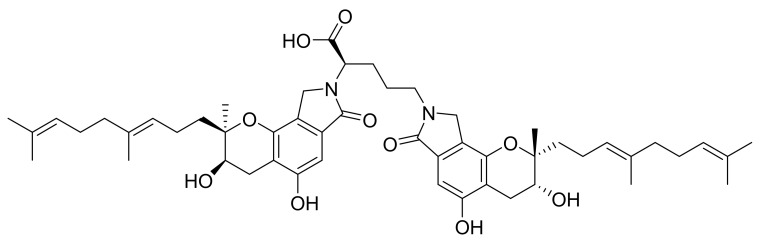
Chemical structure of 2,5-B*H*PA: 2,5-Bis-[8-(4,8-dimethyl-nona-3,7-dienyl)-5,7-dihydroxy-8-methyl-3-keto-1,2,7,8-tertahydro-6*H*-pyran[a]isoindol-2-yl]-pentanoic acid.

**Figure 2 marinedrugs-17-00651-f002:**
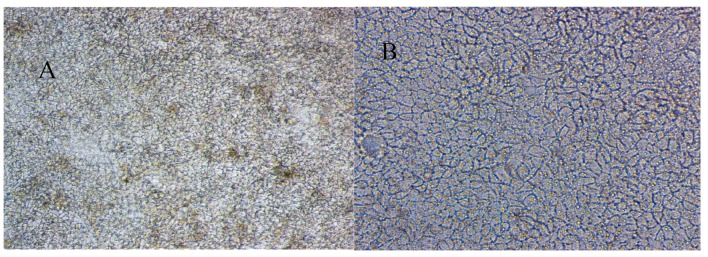
The micrograph of Caco-2 cells monolayer (**A**). 100×: Eyepiece 10×, Objective 10×, Cultured 22d; (**B**). 200×, Eyepiece 10×, Objective 20×, Cultured 22 d.

**Figure 3 marinedrugs-17-00651-f003:**
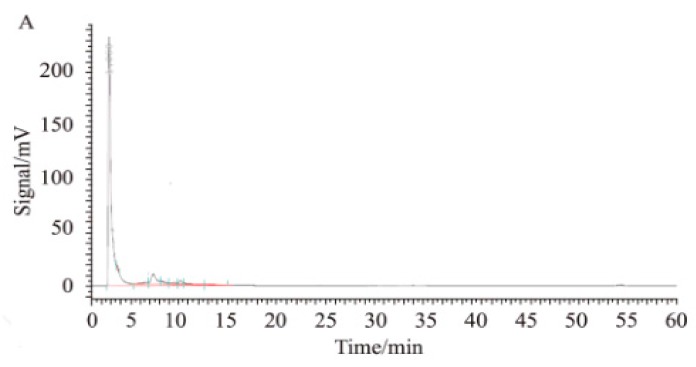
Detection map of 2,5-B*H*PA. (**A**) Blank samples; (**B**) 2,5-B*H*PA standard solution; (**C**) after intravenous injection of biological samples.

**Figure 4 marinedrugs-17-00651-f004:**
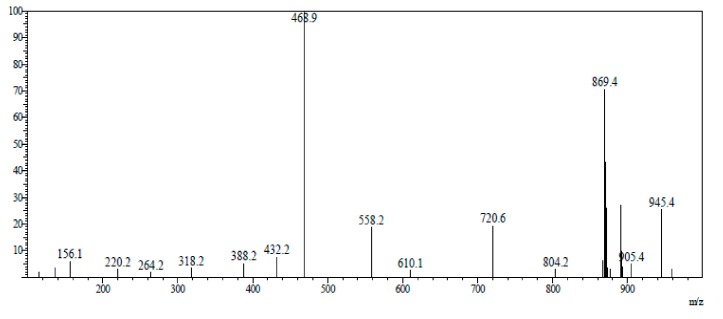
Mass spectrum of 2,5-B*H*PA.

**Figure 5 marinedrugs-17-00651-f005:**
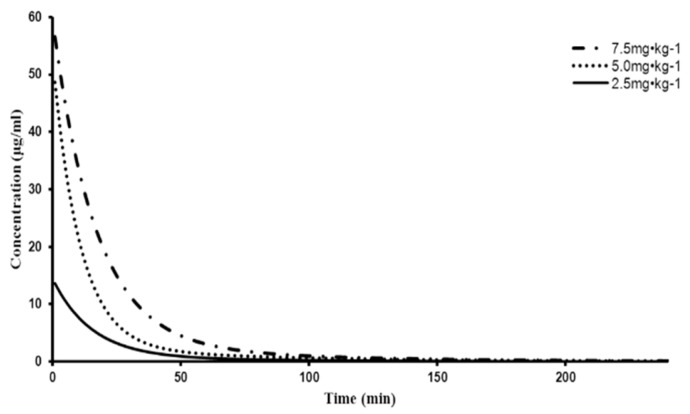
Plasma concentration-time curves of 2,5-B*H*PA after intravenous injection. The values of 7.5, 5.0, and 2.5 mg·kg^−1^ mean that dogs were injected with injected 7.5, 5.0, and 2.5 mg of 2,5-B*H*PA, respectively, per 1 kg of weight.

**Figure 6 marinedrugs-17-00651-f006:**
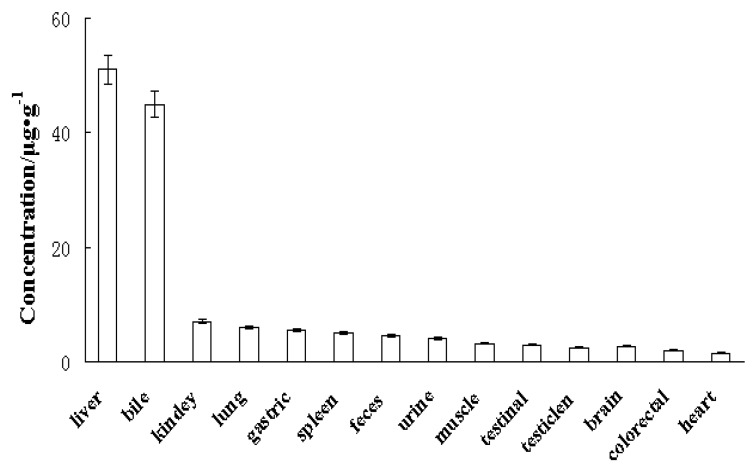
Tissue distribution of 2,5-B*H*PA in dogs (*n* = 3).

**Table 1 marinedrugs-17-00651-t001:** Apparent permeability coefficients (P_app_), efflux ratio, and mean total recoveries of 2,5-B*H*PA (mean ± SD, *n*
*(*Parallel number of sets) = 3).

Compounds	Concentration/ μM	Mean P_app_/10^−6^ cm·s^−1^		Efflux Ratio	Recoveries/%		Total Recoveries/%	
		AP to BL	BL to AP		AP to BL	BL to AP	AP to BL	BL to AP
Fenoterol	2	0.055 ± 0.008	/	/	87.67 ± 1.24	/	90.92 ± 4.23	/
Propranolol	2	26.406 ± 0.854	/	/	92.15 ± 2.42	/	110.01 ± 5.21	/
Digoxin	2	0.033 ± 0.003	9.05 ± 0.62	260	83.66 ± 3.14	82.87 ± 4.38	86.44 ± 3.51	83.31 ± 2.40
2,5-B*H*PA	0.5	0.493 ± 0.016	0.555 ± 0.09	1.13	8.745 ± 0.155	25.54 ± 1.55	35.77 ± 1.29	25.48 ± 1.13
	2	0.048 ± 0.006	0.065 ± 0.005	1.55	13.419 ± 0.566	47.16 ± 1.32	39.67 ± 1.47	47.69 ± 2.10
	5	0.044 ± 0.003	0.043 ± 0.004	1.06	26.76 ± 1.02	52.42 ± 2.20	52.34 ± 4.34	53.02 ± 2.01

Note: All RSD (relative standard deviation) were less 3%. The data represent statistical significance (*p* < 0.05). Apical side (AP); Basolateral side (BL).

**Table 2 marinedrugs-17-00651-t002:** Apparent permeability coefficients (P_app_), efflux ratio, and mean total recoveries of 2,5-B*H*PA after added 0.1% bull serum albumin (mean ± SD, *n* = 3).

Compounds	Concentration/ µM	Mean P_app_/10^−6^ cm·s^−1^		Efflux Ratio	Recoveries/%		Total Recoveries/%	
		AP to BL	BL to AP		AP to BL	BL to AP	AP to BL	BL to AP
Fenoterol	2	0.091 ± 0.012	/	/	94.38 ± 5.14	/	96.16 ± 6.72	/
Propranolol	2	18.12 ± 1.02	/	/	80.32 ± 3.50	/	92.24 ± 3.64	/
Digoxin	2	0.023 ± 0.001	10.351 ± 0.911	484.58	84.14 ± 3.13	93.35 ± 3.56	85.37 ± 3.52	93.43 ± 4.63
2,5-B*H*PA	5	0.097 ± 0.005	0.153 ± 0.012	1.56	85.52 ± 2.34	96.32 ± 5.15	92.83 ± 3.44	96.41 ± 3.52
	15	0.045 ± 0.002	0.132 ± 0.035	2.91	84.85 ± 3.36	96.18 ± 6.56	92.83 ± 4.26	96.97 ± 4.32
	25	0.146 ± 0.009	0.216 ± 0.031	1.49	82.25 ± 2.28	89.52 ± 4.19	90.91 ± 4.63	89.63 ± 3.26

Note: All RSD (relative standard deviation) were less 3%. The data represent statistical significance (*p* < 0.05). Apical side (AP); Basolateral side (BL).

**Table 3 marinedrugs-17-00651-t003:** Recovery and stability of 2,5-B*H*PA in beagle dog plasma (mean ± SD, *n* = 5).

Concentration/μg·mL^−1^	Recovery/%	RSD/%	Stability/%	RSD/%
1.0	98.36 ± 6.38	6.49	94.69 ± 5.37	5.67
100	102.26 ± 4.69	4.59	98.39 ± 2.25	2.29
400	96.58 ± 5.61	5.81	97.63 ± 5.91	6.05

Note: The data represent statistical significance (*p* < 0.05). RSD (relative standard deviation).

**Table 4 marinedrugs-17-00651-t004:** Pharmacokinetics of 2,5-B*H*PA after intravenous injection of 2,5-B*H*PA (mean ± SD, *n* = 3).

Parameters	7.5 mg·kg^−1^	5.0 mg·kg^−1^	2.5 mg·kg^−1^
*K*_10_ (min^−1^)	0.051 ± 0.015	0.075 ± 0.012	0.055 ± 0.021
*K*_12_ (min^−1^)	0.007 ± 0.004	0.017 ± 0.002	0.009 ± 0.004
*K*_21_ (min^−1^)	0.017 ± 0.003	0.018 ± 0.001	0.018 ± 0.001
*T*_1/2_ (min)	49 ± 2	48 ± 2	49 ± 2
AUC_0-t_ (μg·(mL·min)^−1^	1180.5 ± 49.1	717.2 ± 23.6	268.6 ± 19.3
AUC_0-inf_ (μg·(mL·min)^−1^)	1189.0 ± 58.2	723.3 ± 14.78	270.8 ± 35.8
*C*_max_(μg·mL^−1^)	56.48 ± 6.23	48.63 ± 5.53	13.64 ± 2.76
MRT (min)	28.17 ± 1.16	26.23 ± 0.35	28.66 ± 0.84
CL (L·min^−1^·kg^−1^)	0.0062 ± 0.0004	0.0071 ± 0.0008	0.0092 ± 0.0006

Note: The data represent statistical significance (*p* < 0.05). (K: absorption rate constant; AUC: area under the curve; T_1/2_: half-lives; C_max_: maximum concentration; MRT: mean residence time; CL: systemic clearance).

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
