# Peer review of "Evaluation of Marine Diindolinonepyrane in Vitro and in Vivo: Permeability Characterization in Caco-2 Cells Monolayer and Pharmacokinetic Properties in Beagle Dogs"

_marinedrugs, 2019, doi:10.3390/md17120651_

Round 1
Reviewer 1 Report
This paper presents a study on the parmacokinetics of an antithrombotic isoindolone derivative in Beaggle dogs and in a Caco cell model.
The general plan of the paper and the wrting is correct although there are a number of english errors.
However the presentation of the results and the analytical part are not well presented and not well explained.
I noted that the mass spectrometer used is a API 400 . This instrument is made by Sciex not Shimadzu, however the HPLC LC10 is from Shimadzu.
The conditions of the HPLC used in UV and in MS/MS should be fully stated. Two HPLC column are mentionned : one CN column and a Symmetry Waters column.
Which one is used in each assay ?
The m/z values given for the analyzed compounds are sometimes in ESI plus, sometimes in ESI minus. As below :
Paragraph 4.2.1 : You must precise which is ESI + and which is ESI -
Tolbutamide is in ESI plus (M+H+)
Propanolol is also ESI + (M+H+)
Fenoterol is ESI + (M+H+)
Digoxin is probably ESI - (M-H) and
2,5-BHPA also probably ESI -
Please also give the MS/MS parameter. Here you have only given the m/z of each compound. These part must be expanded. You should also state the precision of your calibration curve.
Paragraph 4.2 .2 : You must tell here which column (the symmetry column probably)
It would be interesting here to give a figure if the MS/MS spectrum of 2,5-BHPA
I cannot believe you if you don't show the data and give the fragmentation used in MS/MS
Do you use the software of the API 4000? this should be mentionned.
Line 228 : Tolbutamide (internal standard) was weighted...
Line 268 : HPLC
Line 196 : This line is a repetition of line 190-191
Concerning the whole text and the tables ; problem with the number of significant digits : Please don't give more than 3 digits since all the other are not relevant. Your error is always more that 5%.
Concerning the experimental analytical method ; you describe a HPLC MS/MS method for quantifying a compound for which there is no method described yet (I believe). Thus you should at least show a MS/MS spectum and give the mode (ESI plus or minus) and the ion values selected (m/z of the ion and of the fragment) in the MS/MS procedure and all the parameters used on the mass spectrometer. You should also give a chromatogram using the assay for blank and for instance plasma. Otherwise how can I believe your calibration data ? This could be put in a supplementary document if you think it is too long.
It looks to me that you monitor your compound in ESI minus but the internal standard (tolbutamide) in ESI plus.
Thus as you may understand the paper is not ready for publication. All these imprecisions must be corrected before acceptation.
Author Response
Dear editors,
We sincerely express our thankfulness to the Editor and all the reviewers for their great effort to improve our manuscript quality. The reviewers’ comments give us a big assistance in order to improve our research knowledge and skills. The manuscript has been revised accordingly, and the detailed corrections are listed below point by point:
Reviewer(s)’ Comments to Author:
Reviewer: 1
Comments to the Author
Point 1: However the presentation of the results and the analytical part are not well presented and not well explained.
Response 1: The presentation of the results and the analytical parts are now revised and well presented.
Point 2: I noted that the mass spectrometer used is a API 400. This instrument is made by Sciex not Shimadzu, however the HPLC LC10 is from Shimadzu.
Response 2: It has been revised as per reviewer’s suggestion.
Point 3: The conditions of the HPLC used in UV and in MS/MS should be fully stated. Two HPLC columns are mentioned : one CN column and a Symmetry Waters column. Which one is used in each assay ?
Response 3: The condition of HPLC in UV was varying ratios of acetonitrile-water (0.1% trifluoroacetic acid) and the gradient was from 45:55 to 85:15 (v/v) at a flow rate of 1 mL/min. The column temperature was maintained at 40℃. Eluents were monitored at 265 nm. The condition in MS/MS was varying ratios of acetonitrile-water (0.1% formic acid) at a flow rate of 1mL/min. Electro spray ionization ion source (ESI) was used for anion mode detection. The range of m/z was from 100 to 900. Capillary voltage was 3.0 kV. Taper hole voltage was 40 kV. Ion source temperature was 100℃. Desolvation temperature was 450℃. Velocity of dissolvent gas was 900 L•h-1. The cone gas flow was 50 L•h-1.
The Symmetry C18 5.0 µm (4.6 × 250 mm ) Column of Waters (USA) was used in each assay.
Point 4: The m/z values given for the analyzed compounds are sometimes in ESI plus, sometimes in ESI minus. As below :
Paragraph 4.2.1 : You must precise which is ESI + and which is ESI -
Tolbutamide is in ESI plus (M+H+)
Propanolol is also ESI + (M+H+)
Fenoterol is ESI + (M+H+)
Digoxin is probably ESI - (M-H) and
2,5-BHPA also probably ESI -
Please also give the MS/MS parameter. Here you have only given the m/z of each compound. These parts must be expanded. You should also state the precision of your calibration curve.
Response 4: In the experiment, anion mode (ESI -) was used.
It has been revised as per reviewer’s suggestion. The MS/MS parameter has been expanded in the article and the precision of calibration curve has been stated.
Point 5: Paragraph 4.2 .2 : You must tell here which column (the symmetry column probably)
Response 5: The symmetry column was used.
Point 6: It would be interesting here to give a figure if the MS/MS spectrum of 2, 5-BHPA, I cannot believe you if you don't show the data and give the fragmentation used in MS/MS
Response 6: It has been revised as per reviewer’s suggestion. The MS/MS spectrum of 2, 5-BHPA has been added in article.
Point 7: Do you use the software of the API 4000? This should be mentioned.
Response 7: Yes, It has been revised as per reviewer’s suggestion.
Point 8: Line 228: Tolbutamide (internal standard) was weighted...
Response 8: It has been revised as per reviewer’s suggestion.
Point 9: Line 268: HPLC
Response 9: It has been revised as per reviewer’s suggestion.
Point 10: Line 196: This line is a repetition of line 190-191
Response 10: It has been revised as per reviewer’s suggestion. Line 196 has been deleted.
Point 11: Concerning the whole text and the tables; problem with the number of significant digits: Please don't give more than 3 digits since all the other are not relevant. Your error is always more than 5%.
Response 11: It has been revised as per reviewer’s suggestion. The data have been checked.
Point 12: Concerning the experimental analytical method; you describe a HPLC MS/MS method for quantifying a compound for which there is no method described yet (I believe). Thus you should at least show a MS/MS spectum and give the mode (ESI plus or minus) and the ion values selected (m/z of the ion and of the fragment) in the MS/MS procedure and all the parameters used on the mass spectrometer. You should also give a chromatogram using the assay for blank and for instance plasma. Otherwise how can I believe your calibration data? This could be put in a supplementary document if you think it is too long.
Response 12: It has been revised as per reviewer’s suggestion. The mentioned data have been added in the article as Figure.
Point 13: It looks to me that you monitor your compound in ESI minus but the internal standard (tolbutamide) in ESI plus.
Response 13: No, all are in the ESI minus.
The revised manuscript has been resubmitted to the journal. We are looking forward to the positive response.
Yours sincerely,
Zibin Ma, Ruihua Guo, Jeevithan Elango, Bin Bao and Wenhui Wu
Reviewer 2 Report
The preset work is not sufficient to meet the high standard of this journal. The followings have to be carefully considered.
Line 59, it should be nonsense that the short half-life of 22 min is attributed to a rapid distribution into tissues. An experiment with P-gp overexpressed Caco-2 cells should be involved to judge whether the substance is a substrate of the transporter. Pharmacokinetic parameters have to be recalculated correctly. Figure 3 should be illustrated in a semi-log scale. The authors should calculate and discuss why volume of distributions are different at each dose. Clearances at each dose are not correct, and seem to be significantly decreased by increasing doses. Significant digits for all parameters have to be checked. In Table 4, statistics has to be double-checked. What is ‘survival square sum’? you mean 'sum of squared residuals'? A tissue sampling at 1 h after administration should not be sufficient to represent the tissue distribution, and the time should be justified as well. The instruments should be clearly described for each measurement. The condition of LC-MS/MS should be presented correctly, e.g., mass transition. The preparation for intravenous administration should be described in detail. An internal standard did not seem to be used. What was the evidence of enterohepatic circulation in conclusions?Author Response
Dear editors,
We sincerely express our thankfulness to the Editor and all the reviewers for their great effort to improve our manuscript quality. The reviewers’ comments give us a big assistance in order to improve our research knowledge and skills. The manuscript has been revised accordingly, and the detailed corrections are listed below point by point:
Reviewer(s)’ Comments to Author:
Reviewer: 2
Point 1: Line 59, it should be nonsense that the short half-life of 22 min is attributed to a rapid distribution into tissues.
Response 1: Yes, it has changed: The half-life in blood of 2, 5-BHPA was ~22 min in rats.
Point 2: An experiment with P-gp over expressed Caco-2 cells should be involved to judge whether the substance is a substrate of the transporter.
Response 2: Yes. In the experiment, we used the P-gp to observe whether it is related to the recovery and permeability of the compound 2, 5-BHPA.
Point 3: Pharmacokinetic parameters have to be recalculated correctly.
Response 3: It has been revised as per reviewer’s suggestion. We had recalculated.
Point 4: Figure 3 should be illustrated in a semi-log scale.
Response 4: The concentration-time curves are all illustrated like figure 3.
Point 5: The authors should calculate and discuss why volume of distributions are different at each dose.
Response 5: It has been revised as per reviewer’s suggestion. It has been calculated.
Point 6: Clearances at each dose are not correct, and seem to be significantly decreased by increasing doses.
Response 6: Yes, the doses in the article is from 7.5 mg•kg-1 to 2.5 mg•kg-1. Systemic clearance (CL) values were 0.0062±0.0004, 0.0071±0.0008, 0.0092±0.0006 L•min-1•kg-1 at 7.5, 5.0, 2.5 mg•kg-1, respectively.
Point 7: Significant digits for all parameters have to be checked.
Response 7: It has been revised as per reviewer’s suggestion. We had checked again.
Point 8: In Table 4, statistics has to be double-checked.
Response 8: It has been revised as per reviewer’s suggestion. We had checked again.
Point 9: What is ‘survival square sum’? you mean 'sum of squared residuals'?
Response 9: No. The “survival square sum” is a parameter for the two-compartment model (w = 1/c2).
Point 10: A tissue sampling at 1 h after administration should not be sufficient to represent the tissue distribution, and the time should be justified as well.
Response 10: All the animal tissue distribution experiment chose the time 1 h. So the time 1 h can represent the tissue distribution.
Point 11: The instruments should be clearly described for each measurement.
Response 11: It has been revised as per reviewer’s suggestion.
Point 12: The condition of LC-MS/MS should be presented correctly, e.g., mass transition.
Response 12: It has been revised as per reviewer’s suggestion. It has been checked.
Point 13: The preparation for intravenous administration should be described in detail.
Response 13: It has been revised as per reviewer’s suggestion.
Subcutaneous injection of the forearm was used in the experiment. The injection needle is inserted into the vein at one third of the wrist joint. When the needle is confirmed in the blood vessel, blood flow is seen at the connection tube of the needle. After the injection is completed, press the eye of the needle with a dry cotton ball, quickly pull out the needle, and press locally for a moment to prevent bleeding.
Point 14: An internal standard did not seem to be used.
Response 14: Line 237: Tolbutamide (internal standard) was weighed precisely and dissolved in DMSO as a solvent (0.5 mM for stock solution).
Point 15: What was the evidence of enterohepatic circulation in conclusions?
Response 15: In the discussion, after for 60 min, both liver and small intestine had higher concentration of 2, 5-BHPA than other organs. Therefore, we guessed that 2, 5-BHPA may have the enterohepatic circulation.
The revised manuscript has been resubmitted to the journal. We are looking forward to the positive response.
Yours sincerely,
Zibin Ma, Ruihua Guo, Jeevithan Elango, Bin Bao and Wenhui Wu
Reviewer 3 Report
Zibin Ma et al have evaluated permeability of diindolinonepyrane in Caco-2 cells monolayer and pharmacokinetic properties in beagle dogs. I would suggest next recommendation for improving of paper:
In abstract (and than in the text): I don't see rationality for 3 decimal points for T1/2. In Introduction: authors provide information that 'Caco-2 cell line was derived from the human colon adenocarcinoma cell lines'. How this cell line is linked with thrombosis? Please provide full name for compound in the legend (Fig.1). Tables 1 and 2: please provide the explanation of all abbreviations used in the tables. Please clarify what is mind x+/-s (s=SD or SEM?). Section 2.3: linearity was in ugxmL-1 (lines 106-107), but in Fig. 3 units are mgxmL-1. Please unify units used. In Fig. 3: please indicate what is 7.5 (5.0 and 2.5) mgxkg-1? For better understanding of results I would suggest to present results from lines 121-126 in separate table. Table 4: please explain all abbreviations used. Please indicate +/- SEM or SD? In Section 4.1: please describe shortly how 2,5-BHPA was isolated?Author Response
Dear editors,
We sincerely express our thankfulness to the Editor and all the reviewers for their great effort to improve our manuscript quality. The reviewers’ comments give us a big assistance in order to improve our research knowledge and skills. The manuscript has been revised accordingly, and the detailed corrections are listed below point by point:
Reviewer(s)’ Comments to Author:
Reviewer: 2
Point 1: Line 59, it should be nonsense that the short half-life of 22 min is attributed to a rapid distribution into tissues.
Response 1: Yes, it has changed: The half-life in blood of 2, 5-BHPA was ~22 min in rats.
Point 2: An experiment with P-gp over expressed Caco-2 cells should be involved to judge whether the substance is a substrate of the transporter.
Response 2: Yes. In the experiment, we used the P-gp to observe whether it is related to the recovery and permeability of the compound 2, 5-BHPA.
Point 3: Pharmacokinetic parameters have to be recalculated correctly.
Response 3: It has been revised as per reviewer’s suggestion. We had recalculated.
Point 4: Figure 3 should be illustrated in a semi-log scale.
Response 4: The concentration-time curves are all illustrated like figure 3.
Point 5: The authors should calculate and discuss why volume of distributions are different at each dose.
Response 5: It has been revised as per reviewer’s suggestion. It has been calculated.
Point 6: Clearances at each dose are not correct, and seem to be significantly decreased by increasing doses.
Response 6: Yes, the doses in the article is from 7.5 mg•kg-1 to 2.5 mg•kg-1. Systemic clearance (CL) values were 0.0062±0.0004, 0.0071±0.0008, 0.0092±0.0006 L•min-1•kg-1 at 7.5, 5.0, 2.5 mg•kg-1, respectively.
Point 7: Significant digits for all parameters have to be checked.
Response 7: It has been revised as per reviewer’s suggestion. We had checked again.
Point 8: In Table 4, statistics has to be double-checked.
Response 8: It has been revised as per reviewer’s suggestion. We had checked again.
Point 9: What is ‘survival square sum’? you mean 'sum of squared residuals'?
Response 9: No. The “survival square sum” is a parameter for the two-compartment model (w = 1/c2).
Point 10: A tissue sampling at 1 h after administration should not be sufficient to represent the tissue distribution, and the time should be justified as well.
Response 10: All the animal tissue distribution experiment chose the time 1 h. So the time 1 h can represent the tissue distribution.
Point 11: The instruments should be clearly described for each measurement.
Response 11: It has been revised as per reviewer’s suggestion.
Point 12: The condition of LC-MS/MS should be presented correctly, e.g., mass transition.
Response 12: It has been revised as per reviewer’s suggestion. It has been checked.
Point 13: The preparation for intravenous administration should be described in detail.
Response 13: It has been revised as per reviewer’s suggestion.
Subcutaneous injection of the forearm was used in the experiment. The injection needle is inserted into the vein at one third of the wrist joint. When the needle is confirmed in the blood vessel, blood flow is seen at the connection tube of the needle. After the injection is completed, press the eye of the needle with a dry cotton ball, quickly pull out the needle, and press locally for a moment to prevent bleeding.
Point 14: An internal standard did not seem to be used.
Response 14: Line 237: Tolbutamide (internal standard) was weighed precisely and dissolved in DMSO as a solvent (0.5 mM for stock solution).
Point 15: What was the evidence of enterohepatic circulation in conclusions?
Response 15: In the discussion, after for 60 min, both liver and small intestine had higher concentration of 2, 5-BHPA than other organs. Therefore, we guessed that 2, 5-BHPA may have the enterohepatic circulation.
The revised manuscript has been resubmitted to the journal. We are looking forward to the positive response.
Yours sincerely,
Zibin Ma, Ruihua Guo, Jeevithan Elango, Bin Bao and Wenhui Wu
Round 2
Reviewer 1 Report
The new version does not convince me.
The authors did not take into full account my remarks.
There is an improvement with the added figure 3 and 4.
First, throughout the text and the tables they use number with 4 to 5 significant digits when their analytical methods have an error of at least 5%. This is not correct. They should use a maximum of three digits. Thus please correct everywhere.
They should completely rewrite the experimental concerning the mass spectra :
It is clear that the principal author is not familiar with mass spectrometry. Thus ask your mass spectrometry specialist.
You say that you are using MS/MS (a good possibility with the API4000. However from what you show you are probably only monitoring the molecular ion, not doing MS/MS analysis. Please check with your MS specialist.
I believe that most of the mass spectra molecular ions in positive ESI except digoxin.
Fenoterol C17H21NO4 : M= 303.3 thus if you look at m/z= 304 this is M+H thus ESI+
Digoxin C41H64O14 : M = 780,4 thus if you use m/z= 779.6 you probably use ESI- (M-H)
Propanolol : C16H21NO2 : M= 269.2 thus if you use m/z= 270, you are in ESI+
Tolbutamide : C12H18N2O3S : M = 270,3. Thus if you use m/z= 271, your are in positive ESI (ESI +)
The exact mass of 25-BHPA is Chemical Formula: C51H68N2O10
Exact Mass: 868,4874
Thus in positive mode the m/z should be 869,4 (M+H+) and negative mode ; 867,4 (M-H+). The peak around 891 is probably the M+Na+
You did not understand me when I asked which transition you were looking at. May be you use simply MS not MS/MS.
The big fragment at 468.9 is probably loss of water and loss of one isoindolinone
One remark ; your correction line 93-94 would be better if you said : As shown in figure 3 the compounds present in biological samples are well separated. Figure 4 shows the mass spectrum of 2,5-BHPA in positive ESI.
Thus I am not satisfied from this revision and ask the authors to do it again.
Author Response
Dear editors,
We sincerely express our thankfulness to the Editor and all the reviewers for their great effort to improve our manuscript quality. The reviewers’ comments give us a big assistance in order to improve our research enlightenment and advance. The manuscript has been revised accordingly, and the detailed corrections are listed below point by point.
Reviewer(s)’ Comments to Author:
Reviewer: 1
Comments to the Author
Point 1: First, throughout the text and the tables they use number with 4 to 5 significant digits when their analytical methods have an error of at least 5%. This is not correct. They should use a maximum of three digits. Thus please correct everywhere.
Response 1: It has been revised as per reviewer’s suggestion. The significant digits have been correct with 3 digits in the manuscript everywhere.
Point 2: They should completely rewrite the experimental concerning the mass spectra:
It is clear that the principal author is not familiar with mass spectrometry. Thus ask your mass spectrometry specialist.
You say that you are using MS/MS (a good possibility with the API4000. However from what you show you are probably only monitoring the molecular ion, not doing MS/MS analysis. Please check with your MS specialist.
I believe that most of the mass spectra molecular ions in positive ESI except digoxin.
Fenoterol C17H21NO4: M= 303.3 thus if you look at m/z= 304 this is M+H thus ESI+
Digoxin C41H64O14: M = 780.4 thus if you use m/z= 779.6 you probably use ESI- (M-H)
Propanolol: C16H21NO2: M= 269.2 thus if you use m/z= 270, you are in ESI+
Tolbutamide: C12H18N2O3S: M = 270.3. Thus if you use m/z= 271, your are in positive ESI (ESI +)
The exact mass of 25-BHPA is Chemical Formula: C51H68N2O10
Exact Mass: 868.4874
Thus in positive mode the m/z should be 869.4 (M+H+) and negative mode: 867.4 (M-H+). The peak around 891 is probably the M+Na+
You did not understand me when I asked which transition you were looking at. May be you use simply MS not MS/MS.
The big fragment at 468.9 is probably loss of water and loss of one isoindolinone.
Response 2: Thanks for your comments very much. Your comments give me a big help and I learn more from you about the mass spectrum. First time, I am very sorry, I misunderstand your comments. In the experiment, we used the instrument (mass spectrum API400) which is MS/MS, but we only used MS in the detection. In the article, we have rewritten the experimental concerning the mass spectra from Line 213 to Line 222.
Point 3: One remark; your correction line 93-94 would be better if you said: As shown in figure 3 the compounds present in biological samples are well separated. Figure 4 shows the mass spectrum of 2, 5-BHPA in positive ESI.
Response 3: Thanks for your comments. It has been revised as per reviewer’s suggestion. They have been changed in the article from Line 92 to Line 93.
Reviewer 2 Report
Tha authors have to understand the background of each question, and respond scientifically and soundly.
1. Line 59, it should be nonsense that the short half-life of 22 min is attributed to a rapid distribution into tissues.
2. An experiment with P-gp over expressed Caco-2 cells should be involved to judge whether the substance is a substrate of the transporter.
3. Pharmacokinetic parameters have to be recalculated correctly.
4. Figure 3 should be illustrated in a semi-log scale.
5. The authors should calculate and discuss why volume of distributions are different at each dose.
6. Clearances at each dose are not correct, and seem to be significantly decreased by increasing doses.
7. Significant digits for all parameters have to be checked.
8. In Table 4, statistics has to be double-checked.
9. What is ‘survival square sum’? you mean 'sum of squared residuals'?
10. A tissue sampling at 1 h after administration should not be sufficient to represent the tissue distribution, and the time should be justified as well.
11. The instruments should be clearly described for each measurement.
12. The condition of LC-MS/MS should be presented correctly, e.g., mass transition.
13. The preparation for intravenous administration should be described in detail.
14. An internal standard did not seem to be used.
15. What was the evidence of enterohepatic circulation in conclusions?
Author Response
Dear editors,
We sincerely express our thankfulness to the Editor and all the reviewers for their great effort to improve our manuscript quality. The reviewers’ comments give us a big assistance in order to improve our research enlightenment and advance. The manuscript has been revised accordingly, and the detailed corrections are listed below point by point:
Reviewer(s)’ Comments to Author:
Reviewer: 2
Point 1: Line 59, it should be nonsense that the short half-life of 22 min is attributed to a rapid distribution into tissues.
Response 1: Thanks for your comment and pointing out the obvious mistakes in our manuscript. This statement is wrong. We have revised that the formulation was deleted in Line 59.
Point 2: An experiment with P-gp over expressed Caco-2 cells should be involved to judge whether the substance is a substrate of the transporter.
Response 2: Thanks for your presentation.
In Line 106, we have changed the word “indicated” to another word “suggested”. So the sentence is “It was suggested that 2, 5-BHPA was not a substrate of P-gp and did not participate in cell metabolism”.
Point 3: Pharmacokinetic parameters have to be recalculated correctly.
Response 3: Thanks for your comment. We had recalculated. It is correct. We have got the pharmacokinetic parameters based on plug-in components software of PKsolver V2.0.
Point 4: Figure 3 should be illustrated in a semi-log scale.
Response 4: Figure 3 was from PKsolver V2.0. It is similar with some papers (Example 1: Largazole Pharmacokinetics in Rats by LC-MS/MS. Mar. Drugs 2014, 12, 1623-1640; doi: 10.3390/md12031623; Example2: UHPLC-MS/MS method for the quantification of aloin-A in rat plasma and its application to a pharmacokinetic study. Journal of Pharmaceutical and Biomedical Analysis Available online 25 October 2019, 112928.) The figure 3 doesn’t needs to revising because the numerals were all showing in the X or Y axis of coordinates.
Point 5: The authors should calculate and discuss why volume of distributions are different at each dose.
Response 5: Thank for your advance. This discuss was put in our manuscript, as follows:
AUC (area under the cure) has a collection between the concentrations of sample and removing time. The dose was higher which brings a higher drug concentration of blood. The AUC was relationship with the dose of 2, 5-BHPA closing that the treatment of 7.5 mg/kg dose has higher volume with contrast the treatment of 2.5 mg/kg dose has lower volume, and AUC ratio is about 1.5 between adjacent doses. It is suggested that 2, 5-BHPA was a linear dynamic drugs.
Point 6: Clearances at each dose are not correct, and seem to be significantly decreased by increasing doses.
Response 6: Thanks for your comment. We have noticed the problem. But, the clearances are not immediately related to the concentration, it is also related to the AUC: area under the curve.
Point 7: Significant digits for all parameters have to be checked.
Response 7: Thanks for your comments. It has been revised as per reviewer’s suggestion. The significant digits have been correct with 3 digits in the manuscript everywhere.
Point 8: In Table 4, statistics has to be double-checked.
Response 8: It has been revised as per reviewer’s suggestion. We had checked again using the software of PKsolver V2.0.
Point 9: What is ‘survival square sum’? you mean 'sum of squared residuals'?
Response 9: Thanks for your comments. They are the same meaning. The “survival square sum” is always used as a parameter for the two-compartment model (w = 1/c2).
Point 10: A tissue sampling at 1 h after administration should not be sufficient to represent the tissue distribution, and the time should be justified as well.
Response 10: Thanks for your comment. In the article, we detected the concentration of the compound in blood for 15, 30, 40, 60, 120 and 240 min based on pharmacokinetics experiment. From the experimental results, it shows accumulation and metabolism in liver tissue particularly in 60 min. So, in order to show the experimental results, we chose 60 min after administration to represent the tissue distribution.
Point 11: The instruments should be clearly described for each measurement.
Response 11: It has been revised as per reviewer’s suggestion. The instruments were added in Line 224, Line 247, Line 255, Line 272, Line 283, Line 288, Line 305 and Line 313 to be clearly described for each measurement.
Point 12: The condition of LC-MS/MS should be presented correctly, e.g., mass transition.
Response 12: Thanks for your comments. The conditions have been checked again. In the article, we have rewritten the experimental concerning the mass spectra from Line 213 to Line 222.
Point 13: The preparation for intravenous administration should be described in detail.
Response 13: It has been revised as per reviewer’s suggestion. The preparation for intravenous administration has added in the article from Line 295 to Line 299, as follows:
2, 5-BHPA was prepared with NaHCO3, which was agitated overnight on a magnetic stirrer. Then it was diluted to different concentrations, filtered and stored in refrigerator at 4℃. The dose was adjusted according to the body weight of the beagle dogs in a volume of 200µL and 2, 5-BHPA was administered by intravenous injection (7.5, 5.0 and 2.5 mg/kg). Subcutaneous injection of the forearm was used in the experiment.
Point 14: An internal standard did not seem to be used.
Response 14: Thanks for your comment. Yes, internal standard was not used in the pharmacokinetics experiment. The internal standard of pharmacokinetic experiments was once considered as an important step, and structural analogues were selected as the internal standard. Such an internal label can be replaced by a drug stability test (The recovery and stability of the compound 2, 5-BHPA we have detected is from Line 112 to Line 117), especially a test with a clear separation effect that does not need an internal label. For example, Wang, etal. did not use internal standard (Wang, et al. Pharmacokinetics and tissue distribution of eupatilin and its metabolite in rats by an HPLC-MS/MS method. Journal of Pharmaceutical and Biomedical Analysis. 2018, 159: 113–118. DOI:10.1016/j.jpba.2018.06.037). Manda, et al. also did not use internal standard (Manda, et al. Pharmacokinetics and Tissue Distribution of Aegeline after Oral Administration in Mice. Planta. medica. 2019. DOI:10.1055/a-0851-6879). Pozharitskaya, et al. also did not use internal standard (Pozharitskaya, et al. Pharmacokinetic and Tissue Distribution of Fucoidan from Fucus vesiculosus after Oral Administration to Rats. Mar. Drugs 2018, 16, 132; doi: 10.3390/md16040132)
Point 15: What was the evidence of enterohepatic circulation in conclusions?
Response 15: Thanks for your comment and pointing out the obvious mistakes in our manuscript. Because the enterohepatic circulation and the Caco-2 experiment results are contradictory. So I have deleted the guess in the manuscript in Line188 and Line 324.
The revised manuscript has been resubmitted to the journal. We are looking forward to the positive response.
Yours sincerely,
Zibin Ma, Ruihua Guo, Jeevithan Elango, Bin Bao and Wenhui Wu
Reviewer 3 Report
I have not found responses to my questions:
In abstract (and than in the text): I don't see rationality for 3 decimal points for T1/2. In Introduction: authors provide information that 'Caco-2 cell line was derived from the human colon adenocarcinoma cell lines'. How this cell line is linked with thrombosis? Please provide full name for compound in the legend (Fig.1). Tables 1 and 2: please provide the explanation of all abbreviations used in the tables. Please clarify what is mind x+/-s (s=SD or SEM?). Section 2.3: linearity was in ugxmL-1 (lines 106-107), but in Fig. 3 units are mgxmL-1. Please unify units used. In Fig. 3: please indicate what is 7.5 (5.0 and 2.5) mgxkg-1? For better understanding of results I would suggest to present results from lines 121-126 in separate table. Table 4: please explain all abbreviations used. Please indicate +/- SEM or SD? In Section 4.1: please describe shortly how 2,5-BHPA was isolated?
Author Response
Dear editors,
We sincerely express our thankfulness to the Editor and all the reviewers for their great effort to improve our manuscript quality. The reviewers’ comments give us a big assistance in order to improve our research enlightenment and advance. The manuscript has been revised accordingly, and the detailed corrections are listed below point by point:
Reviewer(s)’ Comments to Author:
Reviewer: 3
Point 1: In abstract (and than in the text): I don't see rationality for 3 decimal points for T1/2.
Response 1: Thanks for your comments. It has been revised as per reviewer’s suggestion. It has been correct with 2 significant digits for T1/2 in Line 22-23, Line 130-131 and Table 4.
Point 2: In Introduction: authors provide information that 'Caco-2 cell line was derived from the human colon adenocarcinoma cell lines'. How this cell line is linked with thrombosis?
Response 2: Thanks for your comment. In the experiment, Caco-2 was used to study the permeability characterization of the compound 2, 5-BHPA. Then we confirmed which administration route is better (intravenous administration or oral route). The compound through oral route is mainly absorbed by intestines, and colon adenocarcinoma cell lines are part of intestines.
Point 3: Please provide full name for compound in the legend (Fig.1).
Response 3: It has been revised as per reviewer’s suggestion. We have added the full name under the figure 1 in Line 55-56.
Point 4: Tables 1 and 2: please provide the explanation of all abbreviations used in the tables.
Response 4: It has been revised as per reviewer’s suggestion. We have provided the explanation of all abbreviations from Line 99 to Line 103.
Point 5: Please clarify what is mind x+/-s (s=SD or SEM?).
Response 5: Thanks for your comment. We should clarify all abbreviations. X is the mean value and S=SD (standard deviation). We have provided the explanation of all abbreviations in the article in Line 99, Line 101, Line 118 and Line 139.
Point 6: Section 2.3: linearity was in ugxmL-1 (lines 106-107), but in Fig. 3 units are mgxmL-1. Please unify units used.
Response 6: Thanks for your comment. In fact, we use mg·L-1 in figure 3. ug·mL-1 is equally to mg·L-1. Traditionally, we always use ug·mL-1 in the linearity of standard curve and use mg·L-1 in the concentration-time curves (mg·L-1 is always used as the dosage of administration, which means every 1kg animal weight inject how much mg drugs or 1L blood contain how much mg drugs).
Point 7: In Fig. 3: please indicate what is 7.5 (5.0 and 2.5) mg·kg-1?
Response 7: Thanks for your comment. We have added it from Line 127 to Line 129, as follows:
7.5 mg·kg-1 means every 1 kg of dogs was injected 7.5 mg the compound 2, 5-BHPA;
5.0 mg·kg-1 means every 1 kg of dogs was injected 5.0 mg the compound 2, 5-BHPA;
2.5 mg·kg-1 means every 1 kg of dogs was injected 2.5 mg the compound 2, 5-BHPA.
Point 8: For better understanding of results I would suggest to present results from lines 121-126 in separate table.
Response 8: Thanks for your comment. The data has presented in Table 4.
Point 9: Table 4: please explain all abbreviations used.
Response 9: It has been revised as per reviewer’s suggestion. We have provided the explanation of all abbreviations in the article.
Point 10: Please indicate +/- SEM or SD?
Response 10: Thanks for your comment. In the article, X is the mean value and +/- S=SD (standard deviation). We have provided the explanation of all abbreviations in the article in Line 99, Line 101, Line 118 and Line 139.
Point 11: In Section 4.1: please describe shortly how 2, 5-BHPA was isolated?
Response 11: It has been revised as per reviewer’s suggestion. We have added the method from Line 198 to Line 200, as follows:
The active compounds 2, 5-BHPA with fibrinolytic activity measured by micro-plate reader were isolated from a culture broth and refined with marine fungi FG216 by using semi-preparative HPLC.
The revised manuscript has been resubmitted to the journal. We are looking forward to the positive response.
Yours sincerely,
Zibin Ma, Ruihua Guo, Jeevithan Elango, Bin Bao and Wenhui Wu
Round 3
Reviewer 1 Report
The authors have made most of the asked corrections. However ther are still two sentences to be corrected since they don't accurately explain the experiments.
Line 486: No it is not multiple reaction monitoring which is a MS/MS method . You are loocking at molecular ions, and are not using the fragmentations in the second quad.
It could be mutiple ion monitoring or SIM: single ions monitoring. Look how the maker of the API4000 calls it. Thermo would use SIM.
Line 489 : (english) 779.6 which was detected...
After these changes the paper could be accepted.
Author Response
Dear editors,
We sincerely express our thankfulness to the Editor and all the reviewers for their great effort to improve our manuscript quality. The reviewers’ comments give us a big assistance in order to improve our research enlightenment and advance. The manuscript has been revised accordingly, and the detailed corrections are listed below point by point.
Reviewer(s)’ Comments to Author:
Reviewer: 1
Comments to the Author
Point 1: Line 486: No it is not multiple reaction monitoring which is a MS/MS method . You are loocking at molecular ions, and are not using the fragmentations in the second quad.
It could be mutiple ion monitoring or SIM: single ions monitoring. Look how the maker of the API4000 calls it. Thermo would use SIM.
Response 1: Thanks for your comments very much. Your comments give me a big help and I learn more from you about the mass spectrum. It has been revised as per reviewer’s suggestion. We had looked the maker of the API4000, it is mutiple ion monitoring. It has changed in Line 219.
Point 2: Line 489: (english) 779.6 which was detected...
Response 2: It has been revised as per reviewer’s suggestion. “779.6 which were detected…” has changed to “779.6 which was detected...” in Line 222.
The revised manuscript has been resubmitted to the journal. We are looking forward to the positive response.
Yours sincerely,
Zibin Ma, Ruihua Guo, Jeevithan Elango, Bin Bao and Wenhui Wu
Reviewer 2 Report
Although the previous points have been partly revised, the revision is getting worse.
The authors have to carefully revise the manuscript, soundly and professionally discuss and add some experiments to meet the international standard of this qualified paper.
The reviewer’s opinion was underlined at the end of each point.
Point 1: Line 59, it should be nonsense that the short half-life of 22 min is attributed to a rapid distribution into tissues.
Response 1: Thanks for your comment and pointing out the obvious mistakes in our manuscript. This statement is wrong. We have revised that the formulation was deleted in Line 59.
Although the sentence was deleted, it should be discussed where such a short half-life comes from?
Point 2: An experiment with P-gp over expressed Caco-2 cells should be involved to judge whether the substance is a substrate of the transporter.
Response 2: Thanks for your presentation.
In Line 106, we have changed the word “indicated” to another word “suggested”. So the sentence is “It was suggested that 2, 5-BHPA was not a substrate of P-gp and did not participate in cell metabolism”.
The authors should carry out an experiment with P-gp over expressed Caco-2 cells to suggest whether the substance is a P-gp substrate or not.
Point 3: Pharmacokinetic parameters have to be recalculated correctly.
Response 3: Thanks for your comment. We had recalculated. It is correct. We have got the pharmacokinetic parameters based on plug-in components software of PKsolver V2.0.
The authors should check both model-dependent and model-dependent parameters and their relationships: e.g., the terminal half-lives calculated from K10s would be 9.2 ~ 13.6 min, which has to be compared with ~50 min of T1/2 in Table 4.
Point 4: Figure 3 should be illustrated in a semi-log scale.
Response 4: Figure 3 was from PKsolver V2.0. It is similar with some papers (Example 1: Largazole Pharmacokinetics in Rats by LC-MS/MS. Mar. Drugs 2014, 12, 1623-1640; doi: 10.3390/md12031623; Example2: UHPLC-MS/MS method for the quantification of aloin-A in rat plasma and its application to a pharmacokinetic study. Journal of Pharmaceutical and Biomedical Analysis Available online 25 October 2019, 112928.) The figure 3 doesn’t needs to revising because the numerals were all showing in the X or Y axis of coordinates.
The readers may want to see the terminal shape and the slope of time courses of plasma concentrations at each dose. The insertion of a semi-log graph should not be a burden.
The measured and fitted data should be overlapped in the same graph.
Point 5: The authors should calculate and discuss why volume of distributions are different at each dose.
Response 5: Thank for your advance. This discuss was put in our manuscript, as follows:
AUC (area under the cure) has a collection between the concentrations of sample and removing time. The dose was higher which brings a higher drug concentration of blood. The AUC was relationship with the dose of 2, 5-BHPA closing that the treatment of 7.5 mg/kg dose has higher volume with contrast the treatment of 2.5 mg/kg dose has lower volume, and AUC ratio is about 1.5 between adjacent doses. It is suggested that 2, 5-BHPA was a linear dynamic drugs.
This explanation can be hardly acceptable. Volume of distribution is independent parameter on the dose.
Point 6: Clearances at each dose are not correct, and seem to be significantly decreased by increasing doses.
Response 6: Thanks for your comment. We have noticed the problem. But, the clearances are not immediately related to the concentration, it is also related to the AUC: area under the curve.
The clearance is also independent parameter, which means it should be constant as long as the dose is not high enough to represent a non-linearity.
Point 7: Significant digits for all parameters have to be checked.
Response 7: Thanks for your comments. It has been revised as per reviewer’s suggestion. The significant digits have been correct with 3 digits in the manuscript everywhere.
The significant digits should be based on the accuracy!! All numbers have to be double-checked.
Point 8: In Table 4, statistics has to be double-checked.
Response 8: It has been revised as per reviewer’s suggestion. We had checked again using the software of PKsolver V2.0.
Was there no statistical significance in Cl?
Point 9: What is ‘survival square sum’? you mean 'sum of squared residuals'?
Response 9: Thanks for your comments. They are the same meaning. The “survival square sum” is always used as a parameter for the two-compartment model (w = 1/c2).
It is still not clear. Does it mean a weighting factor?
Point 10: A tissue sampling at 1 h after administration should not be sufficient to represent the tissue distribution, and the time should be justified as well.
Response 10: Thanks for your comment. In the article, we detected the concentration of the compound in blood for 15, 30, 40, 60, 120 and 240 min based on pharmacokinetics experiment. From the experimental results, it shows accumulation and metabolism in liver tissue particularly in 60 min. So, in order to show the experimental results, we chose 60 min after administration to represent the tissue distribution.
This explanation can be hardly understood. The authors should explain why 60 min!!
Point 11: The instruments should be clearly described for each measurement.
Response 11: It has been revised as per reviewer’s suggestion. The instruments were added in Line 224, Line 247, Line 255, Line 272, Line 283, Line 288, Line 305 and Line 313 to be clearly described for each measurement.
Line 460: “Mass spectrum (API 4000) was purchased from Sciex.” Did the authors really purchase ‘mass spectrum’?
Again, The section has to be carefully revised and clearly written.
Point 12: The condition of LC-MS/MS should be presented correctly, e.g., mass transition.
Response 12: Thanks for your comments. The conditions have been checked again. In the article, we have rewritten the experimental concerning the mass spectra from Line 213 to Line 222.
API4000 is an HPLC-MS/MS system. Did the authors use it as a single mass? Did the authors use an isocratic or a gradient program?, if the latter, then it should be clearly shown.
Point 13: The preparation for intravenous administration should be described in detail.
Response 13: It has been revised as per reviewer’s suggestion. The preparation for intravenous administration has added in the article from Line 295 to Line 299, as follows:
2, 5-BHPA was prepared with NaHCO3, which was agitated overnight on a magnetic stirrer. Then it was diluted to different concentrations, filtered and stored in refrigerator at 4℃. The dose was adjusted according to the body weight of the beagle dogs in a volume of 200µL and 2, 5-BHPA was administered by intravenous injection (7.5, 5.0 and 2.5 mg/kg). Subcutaneous injection of the forearm was used in the experiment.
The authors should explain how to prepare the intravenous solution in detail. e.g., the concentration of NaHCO3.
Point 14: An internal standard did not seem to be used.
Response 14: Thanks for your comment. Yes, internal standard was not used in the pharmacokinetics experiment. The internal standard of pharmacokinetic experiments was once considered as an important step, and structural analogues were selected as the internal standard. Such an internal label can be replaced by a drug stability test (The recovery and stability of the compound 2, 5-BHPA we have detected is from Line 112 to Line 117), especially a test with a clear separation effect that does not need an internal label. For example, Wang, etal. did not use internal standard (Wang, et al. Pharmacokinetics and tissue distribution of eupatilin and its metabolite in rats by an HPLC-MS/MS method. Journal of Pharmaceutical and Biomedical Analysis. 2018, 159: 113–118. DOI:10.1016/j.jpba.2018.06.037). Manda, et al. also did not use internal standard (Manda, et al. Pharmacokinetics and Tissue Distribution of Aegeline after Oral Administration in Mice. Planta. medica. 2019. DOI:10.1055/a-0851-6879). Pozharitskaya, et al. also did not use internal standard (Pozharitskaya, et al. Pharmacokinetic and Tissue Distribution of Fucoidan from Fucus vesiculosus after Oral Administration to Rats. Mar. Drugs 2018, 16, 132; doi: 10.3390/md16040132)
Still unclear!!
The authors wrote:
“..Fenoterol, Propanolol, internal standard (Tolbutamide) and 2, 5-BHPA was...”
“Tolbutamide (internal standard) was weighed....”
Author Response
Dear editors,
We sincerely express our thankfulness to the Editor and all the reviewers for their great effort to improve our manuscript quality. The reviewers’ comments give us a big assistance in order to improve our research enlightenment and advance. The manuscript has been revised accordingly, and the detailed corrections are listed below point by point:
Reviewer(s)’ Comments to Author:
Reviewer: 2
Point 1: Line 59, it should be nonsense that the short half-life of 22 min is attributed to a rapid distribution into tissues.
Although the sentence was deleted, it should be discussed where such a short half-life comes from?
Response 1: Thanks for your comment. The half-life of 22 min was detected in rats in our previous study. The reference 14 has cited in the article. (Su, et al. Pharmacokinetics and tissue distribution of a novel marine fibrinolytic compound in Wistar rat following intravenous administrations. J. Chromatogr B. 2013, 942-943, 77-82.)
Point 2: An experiment with P-gp over expressed Caco-2 cells should be involved to judge whether the substance is a substrate of the transporter.
The authors should carry out an experiment with P-gp over expressed Caco-2 cells to suggest whether the substance is a P-gp substrate or not.
Response 2: Thanks for your comments. We had done the experiment in detecting the absorption and transportation characteristic of 2, 5-BHPA in the human Caco-2 cells monolayer. Judgment standard as follows:
Low permeability: Papp ≤ 2.5 (×10-6 cm/s) |
|
Medium permeability: 2.5 < Papp < 10 (×10-6 cm/s) |
|
High permeability: Papp ≥ 10 (×10-6 cm/s) |
|
Substrate: a)Efflux ratio > 2.5 and Papp(B-A)>2.5,it is most likely P-gp substrates; |
|
b)Efflux ratio > 2.5 but Papp(B-A)<2.5,it could be P-gp substrates; |
From table 2, in the three concentrations of 2, 5-BHPA, two are suitable for the judgment standard but another one is not. So, we cannot confirm that whether the compound is the substrate of P-gp. Based on the experimental results, we have changed the word “indicated” to another word “suggested”.
Point 3: Pharmacokinetic parameters have to be recalculated correctly.
The authors should check both model-dependent and model-dependent parameters and their relationships: e.g., the terminal half-lives calculated from K10s would be 9.2 ~ 13.6 min, which has to be compared with ~50 min of T1/2 in Table 4.
Response3: Thanks for your comment. We had checked our experimental data again. It's the fact that we got this result using the software of PKsolver V2.0.
The main pharmacokinetic parameters are T1/2 (half-life) = 0.693/Kt, Kt (absorption rate constant); CL (Clearances) =dose/AUC. AUC (area under the cure) is not necessarily proportional to dose, especially for nonlinear pharmacokinetics. AUC (area under the cure) and Cmax (maximum concentration) are two independent indicators of drug exposure without any direct relationship. Cmax is not necessarily proportional to dose, especially for nonlinear pharmacokinetics.
Point 4: Figure 3 should be illustrated in a semi-log scale.
The readers may want to see the terminal shape and the slope of time courses of plasma concentrations at each dose. The insertion of a semi-log graph should not be a burden.
The measured and fitted data should be overlapped in the same graph.
Response 4: Thanks for your comments. In the figure 3, we mainly want to show the experiment results—the concentration of the compound 2, 5-BHPA in plama was changing with the time. We had illustrated it in a semi-log sale, but we think it cannot intuitively reflect the whole plasma concentration-time curves. We think when the data is very bigger, the semi-log always is used.
It is similar with some papers in Marine Drugs. (Example1:Largazole Pharmacokinetics in Rats by LC-MS/MS. Mar. Drugs 2014, 12, 1623-1640; doi: 10.3390/md12031623; Example2: Pozharitskaya, et al. Pharmacokinetic and Tissue Distribution of Fucoidan from Fucus vesiculosus after Oral Administration to Rats. Mar. Drugs 2018, 16, 132; doi: 10.3390/md16040132).
In order to keep be same with Marine Drugs, the figure 3 doesn’t needs to revising because the numerals were all showing in the X or Y axis of coordinates.
Point 5: The authors should calculate and discuss why volume of distributions are different at each dose.
This explanation can be hardly acceptable. Volume of distribution is independent parameter on the dose.
Response 5: Thanks for your comments. Your opinion has the same meaning as what we have stated. In my article, it is independent parameter on the dose.
AUC (area under the cure) represents the relative accumulation of a drug over a period of time. In table 4, we can gain the value of AUC and discuss the different as follows:
AUC (area under the cure) has a collection between the concentrations of sample and removing time. The dose was higher which brings a higher drug concentration of blood. The AUC was relationship with the dose of 2, 5-BHPA closing that the treatment of 7.5 mg/kg dose has higher volume with contrast the treatment of 2.5 mg/kg dose has lower volume, and AUC ratio is about 1.5 between adjacent doses. It is suggested that 2, 5-BHPA was a linear dynamic drugs.
Point 6: Clearances at each dose are not correct, and seem to be significantly decreased by increasing doses.
The clearance is also independent parameter, which means it should be constant as long as the dose is not high enough to represent a non-linearity.
Response 6: Thanks for your comment. We had checked our experimental data again. It's the fact that we got this result, or maybe it's the specificity of the drug that we need to find out more about it. It is similar with some papers, e.g. Yan, et al. Pharmacokinetics and tissue distribution of crotonoside. Xenobiotica, 2018; 48(1): 28–36. DOI: 10.1080/00498254.2016.1276311.
Point 7: Significant digits for all parameters have to be checked.
The significant digits should be based on the accuracy!! All numbers have to be double-checked.
Response 7: Thanks for your comments. It has been revised as per reviewer’s suggestion. The significant digits have been double-checked and correct based on the accuracy in the article everywhere.
Point 8: In Table 4, statistics has to be double-checked.
Was there no statistical significance in Cl?
Response 8: Thanks for your comment. We had checked our experimental data again using the software of PKsolver V2.0. The data is correct. It's the fact that we got this result, or maybe it's the specificity of the drug that we need to find out more about it. Others also get the same results, e.g. Yan, et al. Pharmacokinetics and tissue distribution of crotonoside. Xenobiotica, 2018; 48(1): 28–36. DOI: 10.1080/00498254.2016.1276311.
Point 9: What is ‘survival square sum’? you mean 'sum of squared residuals'?
It is still not clear. Does it mean a weighting factor?
Response 9: Thanks for your comments. Yes, it is a weighting factor. The “survival square sum” is always used as a parameter for the two-compartment model (w = 1/c2).
Point 10: A tissue sampling at 1 h after administration should not be sufficient to represent the tissue distribution, and the time should be justified as well.
This explanation can be hardly understood. The authors should explain why 60 min!!
Response 10: Thanks for your comment.
From pharmacokinetics experimental results, the half-life is about 50 min and it shows a good accumulation and metabolism in liver particularly in 60 min. So, in order to observe the compound near the half-life in the organs, we chose 60 min after administration to represent the tissue distribution.
Point 11: The instruments should be clearly described for each measurement.
Line 460: “Mass spectrum (API 4000) was purchased from Sciex.” Did the authors really purchase ‘mass spectrum’?
Again, The section has to be carefully revised and clearly written.
Response 11: It has been revised as per reviewer’s suggestion. We had checked again and revised carefully.
Point 12: The condition of LC-MS/MS should be presented correctly, e.g., mass transition.
API4000 is an HPLC-MS/MS system. Did the authors use it as a single mass? Did the authors use an isocratic or a gradient program? if the latter, then it should be clearly shown.
Response 12: Thanks for your comments. Yes, we use it as a single mass. We use an isocratic program in the article.
Point 13: The preparation for intravenous administration should be described in detail.
The authors should explain how to prepare the intravenous solution in detail. e.g., the concentration of NaHCO3.
Response 13: Thanks for your comment. We have rewritten again. The preparation for intravenous administration has added in the article from Line 295 to Line 300, as follows:
2, 5-BHPA was dissolved in normal saline with NaHCO3 as a solvent (2, 5-BHPA/NaHCO3, m/m, 1:1), which was agitated overnight on a magnetic stirrer. Then it was diluted to different concentrations, filtered and stored in refrigerator at 4℃. The dose was adjusted according to the body weight of the beagle dogs in a volume of 200µL and 2, 5-BHPA was administered by intravenous injection (7.5, 5.0 and 2.5 mg/kg). Subcutaneous injection of the forearm was used in the experiment.
Point 14: An internal standard did not seem to be used.
Response 14: Thanks for your comment. In the article, it has two experiments.
Tolbutamide as internal standard was used in the permeability characterization in Caco-2 cells monolayer experiment.
Internal standard was not used in the pharmacokinetics experiment. The internal standard of pharmacokinetic experiments was once considered as an important step, and structural analogues were selected as the internal standard. Such an internal label can be replaced by a drug stability test (The recovery and stability of the compound 2, 5-BHPA we have detected is from Line 114 to Line 119), especially a test with a clear separation effect that does not need an internal label. For example, Wang, etal. did not use internal standard (Wang, et al. Pharmacokinetics and tissue distribution of eupatilin and its metabolite in rats by an HPLC-MS/MS method. Journal of Pharmaceutical and Biomedical Analysis. 2018, 159: 113–118. DOI:10.1016/j.jpba.2018.06.037). Manda, et al. also did not use internal standard (Manda, et al. Pharmacokinetics and Tissue Distribution of Aegeline after Oral Administration in Mice. Planta. medica. 2019. DOI:10.1055/a-0851-6879). Pozharitskaya, et al. also did not use internal standard (Pozharitskaya, et al. Pharmacokinetic and Tissue Distribution of Fucoidan from Fucus vesiculosus after Oral Administration to Rats. Mar. Drugs 2018, 16, 132; doi: 10.3390/md16040132)
The revised manuscript has been resubmitted to the journal. We are looking forward to the positive response.
Yours sincerely,
Zibin Ma, Ruihua Guo, Jeevithan Elango, Bin Bao and Wenhui Wu
Reviewer 3 Report
The manuscript was improved, however, some minor revison is still required.
In Abstract the number of decimal points should be the same for parameter and SD ( see: '...the peak concentration (Cmax) were...and 13.6±2.76'; '...tthe mean retention time (MRT) were 28.2±1.16, 26.2±0.35 and 28.7±0.84; check the same thinks in Tables 1 and 2) Tables 1 and 2: please provide the explanation of all abbreviations used in the tables in the legend right after tables. Section 2.3: linearity was in ugxmL-1 (lines 106-107), but in Fig. 3 units are mgxmL-1. Please unify units used. This will help readers to understand your mining. As soon as the data of pharmacokinetic are presented in Table 4, the lines 316-320 could be excluded. You can write only: 'The pharmacokinetic parameters of 2, 5-BHPA are presented in Table 4'). In section 4.1 please describe how 2, 5-BHPA was isolated. The English need moderate revision.Author Response
Dear editors,
We sincerely express our thankfulness to the Editor and all the reviewers for their great effort to improve our manuscript quality. The reviewers’ comments give us a big assistance in order to improve our research enlightenment and advance. The manuscript has been revised accordingly, and the detailed corrections are listed below point by point:
Reviewer(s)’ Comments to Author:
Reviewer: 3
Point 1: In Abstract the number of decimal points should be the same for parameter and SD ( see: '...the peak concentration (Cmax) were...and 13.6±2.76'; '...tthe mean retention time (MRT) were 28.2±1.16, 26.2±0.35 and 28.7±0.84; check the same thinks in Tables 1 and 2)
Response 1: Thanks for your comments. It has been revised as per reviewer’s suggestion. It has been correct in the article and tables.
Point 2: Tables 1 and 2: please provide the explanation of all abbreviations used in the tables in the legend right after tables.
Response 2: Thanks for your comments. It has been revised as per reviewer’s suggestion. We had provided the explanation of all abbreviations used in the tables in Line 100, Line 103, Line 119 and Line 137-138.
Point 3: Section 2.3: linearity was in ugxmL-1 (lines 106-107), but in Fig. 3 units are mgxmL-1. Please unify units used. This will help readers to understand your mining.
Response 3: Thanks for your comment. It has been revised as per reviewer’s suggestion. The units had been unified in the article.
Point 4: As soon as the data of pharmacokinetic are presented in Table 4, the lines 316-320 could be excluded. You can write only: 'The pharmacokinetic parameters of 2, 5-BHPA are presented in Table 4').
Response 4: Thanks for your comment. It has been revised as per reviewer’s suggestion. We had changed the lines to “The pharmacokinetic parameters of 2, 5-BHPA are presented in Table 4”.
Point 5: In section 4.1 please describe how 2, 5-BHPA was isolated. The English need moderate revision.
Response 5: Thanks for your comment. It has been revised as per reviewer’s suggestion. We had rewritten it again in Line 194-201 as follows:
The active compounds 2, 5-BHPA with fibrinolytic activity measured by micro-plate reader were isolated from a culture broth and refined with marine fungi Stachybotrys longispora FG216 (CCTCCM 2012272) by using semi-preparative HPLC. This material was subjected to preparative HPLC on an Inertsil PREP-ODS column (22.5×250 mm), which was developed at 40℃with gradient elution of acetonitrile and 0.1% trifluoroacetic acid at a rate of 10 mL/min. The fraction containing the fibrinolytic products was evaporated to remove acetonitrile and trifluoroacetic acid after the purified compounds was extracted with ethyl acetate.
The revised manuscript has been resubmitted to the journal. We are looking forward to the positive response.
Yours sincerely,
Zibin Ma, Ruihua Guo, Jeevithan Elango, Bin Bao and Wenhui Wu